Effects of narrow linear clearings on movement and habitat use in a boreal forest mammal community during winter

Pattison Colin A. 1 2 colin.pattison@sait.ca
Catterall Carla P. 1
1 School of Environment and Science, Environmental Futures Research Institute, Griffith University , Nathan, QLD , Australia
2 MacPhail School of Energy, SAIT Polytechnic , Calgary, AB , Canada
Burke Darren
Electronic publication date: 2019 Feb 25
Publication date: 2019
Volume: 7
Electronic Location ID: e6504
Received 2017 Mar 16; Accepted 2019 Jan 22
Copyright: © 2019 Pattison and Catterall
Copyright year: 2019
Copyright holder: Pattison and Catterall
License: This is an open access article distributed under the terms of the Creative Commons Attribution License, which permits unrestricted use, distribution, reproduction and adaptation in any medium and for any purpose provided that it is properly attributed. For attribution, the original author(s), title, publication source (PeerJ) and either DOI or URL of the article must be cited.
License URL: https://creativecommons.org/licenses/by/4.0/

Keywords: Barrier, Fragmentation, Fossil fuel, Functional group, Seismic, Edge effect, Corridor, Mammal

Funding: School of Environment, Griffith University, Australia Griffith University Postgraduate Research Scholarship Funding was provided by the School of Environment, Griffith University, Australia. Colin A. Pattison was supported by a Griffith University Postgraduate Research Scholarship. The funders had no role in study design, data collection and analysis, decision to publish, or preparation of the manuscript.

==============================
Linear clearings for human activities cause internal fragmentation of otherwise intact native forest, with many potential impacts on wildlife. Across a boreal forest region of some 4,000 km2, we investigated how movements and habitat use of ecologically different mammal species are affected by narrow (about eight m) seismic line (SL) clearings associated with fossil fuel extraction, which form extensive networks many kilometers long. We conducted nine repeat snow track surveys during three winters at 14 pairs of one-kilometer transects, each comprising one transect along the SL and a second running perpendicular into adjacent forest. Data for 13 individually-analyzed mammal taxa (species or sets of closely related species) and five mammal groups, categorized based on body size-diet combinations, showed that movements across transects were either unaffected by SL clearings (relative to continuous forest) or restricted only slightly. However, these clearings were favored for linear travel by most species and body size-diet groups (excepting small mammals). The strength of this preference varied in a manner consistent with species’ differing needs to move long distances (associated with their energetic requirements): large predators > large herbivores > mid-sized predators > mid-sized herbivores > small mammals. In terms of overall habitat use, large-bodied predators (e.g., wolves and coyotes) strongly selected SL clearings over forest, medium-sized predators (e.g., mustelids) and medium-sized herbivores (e.g., hares and squirrels) preferred forest, and neither large herbivores nor small mammals had a clear habitat preference. Consequently, there was a net shift in both species and trophic composition within the SL, in favor of large predators and away from medium-sized predators and herbivores. Given the high regional SL density (1.9 km/km2) such shifts are likely to have complex ecological consequences, of currently unknown magnitude.

Introduction

Long and narrow clearings for human activities divide large forest tracts into multiple smaller patches, leading to a range of potential impacts on animals’ spatial behavior, movements and relationships (Forman, 1995; Benitez-Lopez, Alkemade & Verweij, 2010). For example, development associated with fossil fuel extraction is increasingly creating extensive networks of linear clearings such as roads, recreation trails, powerlines, pipelines, and seismic exploration lines (Butt et al., 2013). In particular, ground-active mammals are likely to be strongly impacted, because these linear forest clearings are more open than, and very different from, intact forest vegetation (Forman, 1995; Haddad et al., 2003).

Research into the impacts of linear clearings on terrestrial mammals has largely focused on roads (Trombulak & Frissell, 2000), but the nature of impact potentially differs according to both the type of clearing and the species or ecological characteristics of particular mammals. For example, road clearings are distinguished by high intensities of vehicular movements and of human use, which induce their avoidance by some animals (Coffin, 2007) but any type of linear clearing could present a barrier to crosswise movement, irrespective of the level of human usage (McGregor, Bender & Fahrig, 2008), because some species simply avoid clearings, even in naturally patchy forests (e.g., boreal, Hodson, Fortin & Belanger, 2010). If cleared ground is a barrier to movement, or if areas adjacent to the forest edge are avoided, forests become functionally fragmented (Smith & Smith, 2009) for affected mammals. Indeed, linear forest clearings are commonly expected to impede movements, especially for smaller mammals, due to their smaller home ranges and limited capacity for movement (Oxley, Fenton & Carmody, 1974; Goosem, 2002; Peter et al., 2013). However, some mammal species preferentially use both road verges and non-road linear clearings because they provide either movement corridors (Gese, Dowd & Aubry, 2013; Tigner, Bayne & Boutin, 2014), or particular food resources (Morelli et al., 2014). Such preferential use for travel has been observed in some large predators, and can then have further indirect effects on other mammal species which avoid areas near linear clearings to reduce the risk of predation (Dyer et al., 2001). Given this diversity of potential responses, evidence remains sparse about how various different mammal species within a forest community are impacted by non-road linear clearings.

Species that differ in functional characteristics such as body size and diet could be expected to vary predictably in their type of response to non-road linear clearings. For example, larger bodied mammals move over relatively larger areas to meet their metabolic requirements (Swihart, Slade & Bergstrom, 1988), predators move the greatest distances (Garland, 1983; Gittleman, 1985), and mammals moving between high quality forage and hunting areas in snow-covered areas adapt their movement behavior to decrease energy expenditure (Telfer & Kelsall, 1984; Potapov et al., 2011). Obstacle-free clearings that extend over long distances are well suited for energetically efficient long-distance linear travel. However, previous research has mainly focused on either single species or subsets of functionally similar species (Harmsen et al., 2010; Gese, Dowd & Aubry, 2013; Tigner, Bayne & Boutin, 2014). Assessments and comparisons of responses to non-road linear clearings by different mammals within the same community have been lacking.

This study aims to determine how non-road linear forest clearings affect different species, of varying sizes and trophic levels, that comprise the diverse mammal community characteristic of boreal forest. Within a region where linear seismic clearings (<10 m wide) have been established in extensive networks to explore for underground oil and gas deposits (Pattison et al., 2016), we placed replicated paired transects in forest and along seismic lines (SLs), in which mammal movements and habitat use were assessed in winter using snow track surveys. We test hypotheses that linear clearings differ from forest in any of: crosswise movements, longitudinal travel, and overall usage, for different individual species and for groups of mammals based on body size-diet (body size-diet hereafter) combinations that reflect differences in species’ energetic needs and ecological relationships. We evaluate the possible role of snow supportiveness as a contributor to observed impacts. We also assess the extent to which these response patterns vary among species and body size-diet groups, and whether the relative propensities of different body size-diet groups to travel preferentially along linear clearings corresponds with their expected energy requirements for movement and thus progressively decrease from large predators to large herbivores to mid-sized predators to mid-sized herbivores to small mammals.

Methods

Study area

The boreal forest study region was located in the foothills of the Rocky Mountains, west of Calgary, Alberta, Canada, at 1,150–1,802 m elevation, across an area of 4,022 km2 (50.9–52.1°N, 114.7–115.2°W; see Pattison et al., 2016). Occupying a transition zone between mountains and grassland, the region supports a diverse mammalian fauna comprising 25–30 species that are typically active in winter, including shrews (Insectivora), rodents (Rodentia), lagomorphs (Lagomorpha), ungulates (Artiocactyla), and carnivores (Carnivora; ASRD, 2000). The mean daily temperature in winter is −6 °C, with mean annual snowfall of 186 cm, falling on an average of 6.5 days/month during October-April (Environment Canada, 2009).

The dominant native vegetation type is coniferous forest in which important species (Archibald, Klappstein & Corns, 1996) include pine (Pinus contorta), spruce (Picea glauca, Picea mariana), fir (Abies lasiocarpa, A. balsamea), and tamarack (Larix laricina); also present are some stands of deciduous forest, dominated by poplar (Populus balsamifera), aspen (Populus tremuloides), and birch (Betula papyifera). At the time of the study, landscape cover in the study region was 67% coniferous forest (in which the cover of tree crowns was at least 60%), 4% deciduous forest and 29% non-forest vegetation including cleared former forest with some agricultural areas (Pattison et al., 2016). Some forest areas were in use for commercial timber extraction, and the study region therefore included a mosaic of various successional stages of regeneration. Forested areas were also criss-crossed by SLs and other linear infrastructure clearings connected with exploration for, and extraction of, below-ground oil and gas. Clearings for SLs (henceforth also termed SLs) were about eight m wide and many kilometers long, and formed extensive networks. They occupied about 1.1% of otherwise forested land, at an average density of 1.9 km/km2; the wider linear clearings for powerlines, pipelines and roads, together accounted for 1.8%, and 0.8 km/km2 (Pattison et al., 2016; Fig. 1).

Figure 1 Example of site layout showing a typical landscape context, transect layout and three types of mammal movements recorded by this study.

Example of a study site layout: (A) shows a typical landscape context and the paired seismic line (SL) and forest transects; (B) diagrams the transect details, including their division into 10, 0.1 km sub-transects and the three types of individual movement (approaches, crossings, or travels).

Study design

We surveyed mammals in 14 replicate 1.0 × 1.0 km sites (Alberta, Tourism, Parks & Recreation, approval 11-014, 12-087), dispersed across a study region that extended 142 km north-south and 28 km east-west. Each site contained a pair of transects (SL and forest transects), each partitioned into ten 0.1 km sub-transects (Fig. 1). Site selection was constrained by logistic and safety concerns. Sites were positioned to obtain adequate replication, while also maximizing their: spatial dispersion (mean nearest neighbor distance 6.0 km, SE 0.8; range 3.2–12.1 km); separation from both roads (>100 m) and areas of timber extraction (>500 m); and amount of coniferous forest cover (apart from SL clearings). Average forest cover (mean 98.4%, SE 1.4%) was affected by two sites that contained areas from which timber had been extracted during 1990–2000 (3% and 20% of sites’ areas), where trees had regrown to six m or more in height before this study commenced.

Each SL transect was aligned along a SL and occupied its complete width, averaging 7.7 m (SE 0.4, range 6.3–11.2; n = 14 sites). SLs were inaccessible to conventional motor vehicles, but some had been widened by off-road vehicle activity, while infrequently-used SLs were narrowed by tree and shrub encroachment. Each forest transect was equal in width to its paired SL transect, and was placed approximately perpendicular to it (depending on local terrain), extending into the forest (Fig. 1). Half of the forest transects were oriented north to south and half east to west. Although established in continuous forest as far as possible, eight forest transects were unavoidably crossed by SLs, accounting for 0.9% on average (SE = 0.3%, range 0–4%, n = 14 transects) of their areas.

In the area surrounding site transects (1.5 km in all directions), SL density averaged 2.6 km/km2 (SE 0.3, range 0.9–4.4, n = 14). Other linear clearings (roads, powerlines, pipelines) averaged 1.0 km/km2 (SE 0.2, range 0.3–2.5) in density.

Each transect was surveyed (details below) for mammals and snow conditions nine times over three winters (twice in December 2010–April 2011, three times in December 2011–April 2012, four times in November 2012–April 2013). Surveys of both transects within each site were completed on the same day and generally one or two sites were surveyed per day. Repeat surveys were not conducted until sufficient snow had fallen to completely obliterate tracks recorded in the previous survey. Transects were surveyed by either one (2012–2013) or two observers (2010–2011) travelling on foot.

Mammal surveys

During each survey, all mammal tracks on each sub-transect were classified by trained observers using reference materials (Forrest, 1988), to the finest possible taxon (Table 1); usually to species but sometimes to genus, and for voles to several related genera; in some cases identification involved backtracking up to 100 m. Each taxon was also assigned to one of five body size-diet groups based on combinations of body weight and diet: large predators (>10 kg), large herbivores (>94 kg), mid-sized predators (0.13–9.1 kg), mid-sized herbivores (0.25–1.5 kg), and small mammals (≤20 g; Table 1).

Table 1 Mammals identified in study.

Mammals identified in the study, their functional groups based on diet and size, and relative occurrence (numbers of sites out of 14, and track characteristics).

Taxona	Abra	Speciesa	Weight (kg)b	Functional groupb	No. sites	Total tracksc	Movement distance (m/km)d	
Grizzly bear	gb*	Ursus arctos	204	Large predator	1	1	2.7	
Cougar	cu	Puma concolor	67	Large predator	13	102	14.6	
Gray wolf	gw	Canis lupus	37	Large predator	10	126	41.1	
Coyote	cy	Canis latrans	16	Large predator	14	414	137.7	
Lynx	ly	Lynx canadensis	12	Large predator	13	120	7.7	
Bobcat	bc*	Lynx rufus	9.1	Mid-sized predator	3	4	0.1	
Fisher	fi*	Pekania pennanti	3.5	Mid-sized predator	7	15	0.7	
Red fox	rf*	Vulpes vulpes	5.5	Mid-sized predator	9	19	1.0	
Marten	ma	Martes americana	1.0	Mid-sized predator	13	1,829	55.9	
Weasel	we	Mustela erminea, M. frenata, M. nivalis	0.13	Mid-sized predator	14	383	12.8	
Moose/elk	mo/ek	Alces alces, Cervus elaphus	355	Large herbivore	14	627	27.8	
Deer	de	Odocoileus hemionus, O. virginianus	94	Large herbivore	14	6,949	304.1	
Hare	ha	Lepus americanus	1.5	Mid-sized herbivore	14	8,191	248.1	
Red squirrel	rs	Tamiasciurus hudsonicus	0.25	Mid-sized herbivore	14	14,309	437.6	
Vole	vo	Clethrionomys gapperi, Phenacomys intermedius, Microtus longicaudus, M. pennsylvanicus	0.020	Small mammal	14	1,737	40.2	
Mouse	mu	Peromyscus maniculatus	0.016	Small mammal	14	1,015	27.5	
Shrew	sh	Sorex arcticus, S. cinereus, S. hoyi, S. monticolus	0.005	Small mammal	14	1,275	27.1	
All groups					–	37,116	1,386.7	
Notes:

a Tracks were identified to the finest possible taxon and potential species for the study area were based on Forrest (1988); Abr shows abbreviations used in Figures; asterisks indicate uncommon species that were not individually analyzed.

b Weights obtained from Harestad & Bunnel (1979). Functional groups were based on body weight and diet; “small mammals” combines small herbivores and shrews, as the prey of larger predators.

c Total number of tracks recorded (approaches(A)+crossings(C)+travels(D)).

d Average on-transect track distance (8C+D) m per km surveyed; averaged across 28 transects, each surveyed nine times; the total distance surveyed was 252 km (nine surveys × 14 sites × two transects × one Km).

Each identified track was classified (Fig. 1) as either a crossing, approach or linear travel. Crossings completely spanned a transect width within a longitudinal distance of <10 m, having originated within the adjacent forest. Approaches also originated within adjacent forest, but they stopped and returned to the forest on the entry side at or slightly beyond the transect edge. Linear travels entered a transect either from adjacent forest or in a clearing at one end, and they continued in a straight or diagonal linear path along the transect for at least 10 m (measured parallel to the transect edge) before entering the forest on either side or continuing past the transect end. Linear travel distance (m) was then measured as the net resulting longitudinal displacement (within the transect and parallel to its edge).

For each taxon and each body size-diet group during each of the nine repeat surveys, we calculated the numbers of crossings (C), approaches (A), and the total linear travel distance (D), for each sub-transect (100 m length), and for each transect (1.0 km). Three further measurements of individuals’ responses to the presence of each transect were then calculated. First, crossing propensity was calculated as 100C/(C+A), that is, the percent of direct approaches from the adjacent forest habitat that resulted in a crossing movement. Second, linear travel propensity was calculated as 100D/(D+8C), that is, the along-transect linear travel track length as a percent of the total on-transect track length; taking the length of a crossing movement as 8.0 m (the rounded average value of SL width). Third, the total on-transect track extent (D+8C; m of track/km surveyed) was employed as an indicator of the relative usage of forest or SL habitats, with larger values reflecting the combined effects of higher abundance, preferential use, and travel speed. Habitat usage intensity was also calculated for individual 100 m sub-transects within the forest transects.

Part way through the study, the trees in portions of three sites were harvested, reducing the transect length for six forest and two SL surveys to 800 m and another four SL surveys to 600 m. Data were simulated for missing sub-transects in post-harvest years using integer averages of taxon-specific tracks recorded on these sub-transects pre-harvest; the simulated data comprised about 1% of all analyzed tracks.

Snow depth and support

At each survey, snow depth (SD) and snow penetration (SP) were recorded at the beginning and end of each sub-transect (11 repeats per transect). SP was measured at the crosswise position on the transect where snow was judged to be most supportive by dropping a 20 g weight from a height of 1.0 m and recording its final depth. SD and SP measurements (in meters) were then averaged for each transect, and a snow support index was calculated as 100(SD-SP)/SD, that is, the percent of snow depth that was resistant to the dropped weight; higher values indicate better support for surface-active animals.

Statistical analyses

The study design uses sites as the unit of replication in statistical analyses, and the habitat type (e.g., SL or forest) as the predictor variable; the analyzed response variables are the site-scale measurements of mammal tracks (with each taxon or body size-diet group separately analyzed as an independent response variable) and snow condition. Preliminary comparisons of track distance (m/km surveyed) across the three sampling winters using repeated measures ANOVA, with sites as subjects and year-site combinations as replicates (3 × 14) showed no significant effect of sampling year (P > 0.05) for nine of 13 separately-tested taxa, with only four being significant: gray wolf (P = 0.004), red squirrel (P < 0.001), mouse (P = 0.005), and shrew (P < 0.001). For efficiency in subsequent analyses, irrespective of taxon, we used the average measurements across the nine repeat surveys at each transect for all taxa, consistent with our focus on spatial rather than temporal processes. For large-bodied species, there could be some occasions where large home ranges enabled particular individuals to contribute in part to measurements at more than one site, but an individual’s behavior at a given site is here viewed as an independent response to the environment at that time and place, and the measurements used in analyses were based on all relevant tracks within a taxon or body size-diet group irrespective of individual mammals (being site-scale averages of all the outcomes of all individual’s independent decisions across nine repeat visits during 3 years).

Differences between SL and forest transects in crossing propensity, linear travel propensity and habitat usage intensity were tested using paired t-tests (n = 14 transect pairs; Currell & Dowman, 2009) for all individual taxa and body size-diet groups, for which at least 100 tracks had been recorded from at least 10 sites. In cases where data did not meet the normality assumption tests were also calculated using Wilcoxon signed rank tests for paired samples. Where the homoscedasticity assumption was violated, values were transformed, which also addressed outliers (Tables S1–S3). Mean difference confidence intervals (95%) and Cohen’s effect sizes were calculated (Lakens, 2013) and power to detect mean differences of 20, 50, and 80% (α = 0.05) using t-tests with observed sample variances (Thomas, 1997, but see Hoenig & Heisey, 2001) was assessed using G*Power (Faul & Erdfelder, 1992). Snow depth and support were compared using t-tests and we used linear regression to test the relationships between linear travel propensity and snow support on SL transects (n = 14 sites).

To test the effect of proximity to SL edges on habitat usage intensity within forest transects, sub-transect (each 0.1 km) values for each mammal, were compared using repeated measures ANOVA, with sites as subjects. Repeated measures is a design where subjects are measured repeatedly (Quinn & Keough, 2002). Here, sites were measured repeatedly by sampling sub-transects (10 per site) which were located at varying distances form the nearest SL. Sub-transects were grouped into three distance categories based on this distance: 0–50 m (n = 36 sub-transects), 51–150 m (n = 44), 151–570 m (n = 60). The maximum distance of 570 m and the unequal sample sizes occurred because of SLs that crossed eight forest transects, or when the “far” end of some forest transects was near a SL. Repeated measures ANOVA assumes normality of residuals and equal variances. Sub-transect measures were log-transformed prior to analysis and where Mauchly’s test for sphericity (equal variances) was violated, the Greenhouse-Geisser correction was used. Residuals were assessed visually using box-plots and the Shapiro-Wilk test. Effect size was calculated as partial and generalized eta squared (Lakens, 2013). Analyses were conducted using excel and R (R Core Team, 2014; ez and stats packages). The potential for false discovery by multiple t-tests and ANOVAs was assessed using the Benjamini-Hochberg procedure (Pike, 2011) with a false discovery rate of 0.15.

Results

The 17 identified taxa (Table 1) accounted for 99% of the recorded tracks, and comprised five large predator species, five mid-sized predator taxa, two large herbivore taxa, two mid-sized herbivore species, and three small mammal taxa. The most common taxa in each body size-diet group were, respectively, coyotes, marten, deer, squirrels and voles, and 13 taxa were sufficiently common to be individually analyzed (Table 1). Across all 17 taxa, approaches comprised 8% of all identified tracks. Most (89%) of the remainder (i.e., on-transect tracks) were crossings, the others being linear travels. However crossings contributed less of the recorded on-transect track length (76%) because of the greater individual lengths of linear travel movements. The results presented below have been condensed but additional details, results of power analysis and Benjamini-Hochberg procedure are provided in Tables S1–S11.

Mammal movements and habitat use in SL clearings versus forest

One of the 13 individually-analyzed taxa (hare), and two of the five body size-diet groups (mid-sized predators and large and mid-sized herbivores), crossed SL transects slightly but significantly less often than was the case for continuous forest transects (Figs. 2A and 3A). However, even in these three cases, the reduction in crossing propensity was small as the average percent of approaches to SLs that resulted in crossings was 89%, compared with 92% in forest (Figs. 2A and 3A; Table S1) and for taxa and body-size diet groups, the mean difference detected was 13% or less (|0.4|–|13|%, Table S1). Small mammals (mouse, shrew) trended toward lower crossing rates in forest compared to SLs, although mean differences were not consistently positive or negative across sites which led to confidence intervals which ranged from positive to negative (Table S1).

Figure 2 Comparison of mammal crosswise movement, linear movement and habitat use, between seismic line clearings and forest habitat for different species.

Comparison of mammal movement and habitat use between seismic line (SL) clearings and forest habitat for different species. (A) Crossing propensity (% of transect approaches which fully crossed), (B) linear travel propensity (% of track length which was linear along the transect), (C) habitat usage intensity (total on transect track extent). Species are ordered by body size-diet group (see Table 1 for abbreviations): large predator (L pred), mid-sized predator (M pred), large herbivore (L herb), mid-sized herbivore (M herb), small mammal (S mam). Values are means and standard errors (n = 14 paired transects per habitat), with results declared significant by Benjamini-Hochberg (BH) procedure and paired t-test P-values (unadjusted, *P < 0.05, **P < 0.01, ***P < 0.001). Results declared non-significant by BH procedure and unadjusted P < 0.1 indicated by + (Tables S8–S10).

Figure 3 Comparison of mammal crosswise movement, linear movement and habitat use, between seismic line clearings and forest habitat for functional groups based on body size-diet combinations.

Comparison of mammal movement and habitat use between seismic line (SL) clearings and forest transects for different body size-diet groups: large predator (L pred), mid-sized predator (M pred), large herbivore (L herb), mid-sized herbivore (M herb), small mammal (S mam). (A) Crossing propensity (% of transect approaches which fully crossed), (B) linear travel propensity (% of track length which was linear along the transect), (C) habitat usage intensity (total on transect track extent for multiple species recorded simultaneously). Values are means and standard errors (n = 14 paired transects in forest and SLs), with results declared significant by Benjamini-Hochberg (BH) procedure and paired t-test P-values (unadjusted, *P < 0.05, **P < 0.01, ***P < 0.001). Results declared non-significant by BH procedure and unadjusted P < 0.1 indicated by + (Tables S8–S10).

Seismic lines had substantial impacts on linear travel movements and habitat use, which varied among taxa and body size-diet groups. Five taxa and three body size-diet groups significantly favored SLs for linear travel, compared with forest transects (Figs. 2B and 3B). Across all mammals, linear travels averaged 42% and 6% of the distances moved within SL and forest transects respectively (Fig. 3B; Table S2), even though the total habitat usage intensity (movement in any direction) was similar in SLs and forest (Fig. 3C; Table S3).

Large predators showed particularly strong preferential use of SL clearings in two ways. First, linear travel averaged 92% and 19% of their distance moved in SL and forest transects, respectively (Fig. 3B; Table S2); three of four analyzed component species also showed significant differences (Fig. 2B). Second, the overall habitat usage intensity by large predators, both collectively and individually, was much greater in SLs than in forest (Figs. 2C and 3C; Table S3). Large herbivores collectively preferred SLs for linear travel (averaging 54% and 14% of movement distances in SLs and forest, respectively), together with a statistically significant difference in both component taxa (Figs. 2B and 3B). However, their habitat usage intensity in SLs was collectively similar to that in forest (Fig. 3C), and was significantly greater only for the rarer of the two component taxa (moose/elk; Fig. 2C).

Mid-sized predators used linear travel less frequently, but still favored SLs for this purpose (averaging 15% and 3% of their distances moved in SLs and forest respectively, Table S2). However both mid-sized predators and herbivores had significantly greater usage intensity in forest than in SL clearings (Fig. 3C), indicating a forest preference for other activities. Component taxa within these groups showed no consistent patterns, apart from strong avoidance of SL clearings by marten (Fig. 2C; Table S3). Among small mammals, linear travel movements were uncommon, and there was little difference between SL and forest in either linear travel or usage intensity (Figs. 2B, 2C, 3B and 3C).

Snow depth and support

Snow was deeper on SLs than in forest (respective mean depths 0.22 and 0.17 m; SEs 0.02, 0.01; paired t-test P = 0.002, n = 14 transect pairs), and also more supportive (mean index values 69% and 54%; SEs 1.7, 2.0; P < 0.001). Correlations between linear travel propensity and snow support were significantly positive for large herbivores collectively and for deer (respectively, r2 = 0.33, 0.41, P = 0.03, 0.01; n = 14 SL transects), but not for other taxa or body size-diet groups (all r2 < 0.22, P > 0.05).

Edge effects on mammal activity in forest adjoining SL clearings

Most mammals within forest adjoining SL clearings showed no edge-sensitivity. The on-transect usage intensity was greatest for sub-transects closer to SLs among only the moose/elk taxon which significantly selected for locations nearer to SLs (Fig. 4; Table S4). In contrast, shrew tended to weakly avoid forest sub-transects closest to SL clearings (Fig. 4). ANOVAs for the remaining 11 taxa and five body size-diet groups all gave P-values > 0.10, graphical data visualizations showed no strong trends and effect sizes were generally low (ηp2 < 0.20) for all but moose/elk (ηp2 = 0.451, Table S4).

Figure 4 Effect of proximity to seismic line clearings on forest habitat usage intensity by affected mammals.

Mean moose (A) and shrew (B) habitat usage intensity (meters of track per sub-transect, standard error) within 100 m sub-transects which were located in forest, at various distances from the nearest seismic line. Results of repeated measures ANOVA where forest sub-transects within sites (subjects, N = 14) were measured repeatedly (10 sub-transects per site) and classified by distance to the nearest seismic line (0–50 m (n = 36 sub-transets); 51–150 m (n = 44); 151–570 m (n = 60)), to assess the effect of distance to SL on habitat usage intensity (*P < 0.05). Refer to Table S4 for results for all mammals analyzed.

Discussion

Effects of linear forest clearings on functional habitat fragmentation

Mammal crossing rates of SL clearings were not substantially different from crossing rates on forested control transects for most taxa recorded in this study area. Even the taxon for which SLs had a significant negative effect on crossing propensity, the magnitude of decrease was quantitatively minor (9%) and two small mammals (mouse and shrew) trended toward higher rates of SL crossing compared to forest. These results suggest that SLs were not strong barriers to movement, a finding contrary to the widely expected idea that linear forest clearings substantially increase habitat fragmentation by impeding movements, especially for smaller mammals, which have limited mobility and smaller home ranges (Peter et al., 2013). Indeed, previous studies in temperate (Oxley, Fenton & Carmody, 1974; Rico, Kindlmann & Frantisek, 2007) and tropical forests (Goosem, 2002) found that unsealed roads with widths similar to the SLs in our study region impeded small mammal crossing movements, although this may have been at least in part a response to motor vehicles and road surface features. However other types of non-road linear clearing such as powerline corridors in tropical forest (Goosem, 1997) and ski pistes in temperate forest (Negro et al., 2013) have also impeded small mammal movements.

In general, small mammals are impeded more by wider openings than narrow openings and more by linear clearings covered in materials which differ from the surrounding forest (e.g., asphalt, Peter et al., 2013). The clearing widths shown to have an inhibitory influence on small mammal crossings elsewhere were 60 m (powerline corridor; Goosem, 1997) and 83 m (ski pistes; Negro et al., 2013). In contrast, this study’s SLs were much narrower (eight m). Furthermore, both the SLs and forest in our boreal study were covered in snow to a depth of about 20 cm. Therefore, our study’s finding of little impedance of small mammal crossings by non-road linear clearings may be due to their relatively narrow width together with their winter surface resemblance to the surrounding forest. Our results apply only to surface movements. Small mammals and their predators (e.g., weasel, marten) move in the subnivean space during the snow cover period, and the more compacted snow in the SL clearings could potentially impact such movements. Further research is needed to test this possibility.

It is also possible that the study region’s SL clearings may constitute a greater movement barrier during summer, when the ground surface structure and composition contrast more strongly with surrounding forest. In evergreen Australian woodlands, movements of small mammals were unaffected by very narrow SLs (1.4 and 4.2 m), but some were impeded by wider fire access clearings (six to seven m) which had been surfaced with gravel (Carthew, Jones & Lawes, 2013). To understand the relative importance of clearing width and surface material (and their interaction) requires further comparative research in other regions and seasons; a significant information gap since linear seismic clearings occur extensively throughout forested areas overlying oil and gas deposits (Rabanal et al., 2010; Kolowski & Alonso, 2012; Carthew, Jones & Lawes, 2013; Pattison et al., 2016).

It remains possible that some mammals’ movements may have been restricted at rates which were too small in magnitude to be detected by this study. Although the current study required a large sampling effort, detection of very small differences would require additional sampling. Effect sizes have been provided (Table S1) to support future research planning, however the biological importance of slight restrictions on movement should also be considered in any such future research.

Seismic line clearings also had very little spatial edge effect on mammal habitat use in adjacent forest (at least in winter, and up to 570 m distant; Fig. 4), across diverse taxa and body size-diet groups. Other studies have concluded that altered physical conditions within forest adjacent to clearings can lead to increased abundances of shrubs and saplings (Harper et al., 2005), and these may in turn attract some mammals to the forest edges (Lidicker, 1999), which may explain this study’s observed increase in habitat usage intensity by moose/elk within 50 m of SLs. On the other hand, some species could be expected to avoid these same areas, as either a direct response to the edge conditions or an indirect response to the presence of humans or predators (Dyer et al., 2001). However, we found only one near-significant negative trend for shrews. This result may have been produced due to unequal sample sizes between sites (subjects) which reduced statistical power or if SLs had a strong negative effect on mammal habitat use at all distances up 570 m from their edges. However, we saw very little difference in habitat usage intensity as distance from SLs increased.

In contrast, Dyer et al. (2001) reported that a large herbivore (caribou) avoided forest within 100 m of SLs, interpreted as a learned response to higher abundances of predators such as gray wolves. Caribou are known to avoid predators by spatially segregating from them (Festa-Bianchet et al., 2011). This raises the question of why there was no such avoidance by the large herbivores in our study, but rather an increase in habitat usage near SL edges in the case of moose/elk. Differing species-specific prey escape modes (Wirsing, Cameron & Heithaus, 2010) are a likely contributing factor. Among our study’s large herbivores, elk escape predation by fleeing and moose use active defence. Deer use a combination of fleeing, active defence and early detection. Use of these alternative strategies to predator avoidance could enable this study’s large herbivores to persist in seeking opportunities for travel and resource access associated with non-road linear clearings, albeit with potentially greater risk of encountering predators (McKenzie et al., 2012). Elsewhere, despite avoiding forest within 50 m of linear clearings when resting, elk were more likely to engage in long distance movements in areas with high predation risk in close proximity to linear clearings (Frair et al., 2005).

Human use deterred wolves and elk from using forest habitat within 50 m of recreational trails in another boreal forest region (Rogala et al., 2011). However, human foot tracks on our study SLs averaged only 39 m/km and were restricted to six of 14 sites over all three winters, providing very little opportunity for edge-avoidance responses to humans.

Other influences of linear clearings on mammal activities and communities

It has long been suggested that mammals may frequently use non-road linear forest clearings for movement (Forman, 1995), but supporting empirical evidence has been lacking for many species. Individual-based studies of movement paths using telemetry and back-tracking data have shown that large predators such as gray wolf and coyote move preferentially along forest trails (Whittington, St. Clair & Mercer, 2005; Latham et al., 2011; Gese, Dowd & Aubry, 2013). In the present study, a different method (site-based measurements of snow track data) yielded a similar finding for both of these species, as well as for cougar, the combined moose/elk taxon, deer and collectively for large and medium-sized predators and large herbivores.

Snow supportiveness was greater in this study’s SLs than in surrounding forest, and across sites the positive correlations between SL snow supportiveness and linear travel propensity for deer and large herbivores suggests that this increased support may have contributed to the appeal of SLs as travel routes. However travel movements of large predators collectively were uninfluenced by snow supportiveness. Gese, Dowd & Aubry (2013) concluded that coyotes selected forest trails for their more compacted snow conditions. However the snow depth in that study (0.78 m) had greater potential to impede large carnivores’ movements than was the case for the 0.17 m deep snow in the present study’s forest transects. Our SLs were also likely favored for linear movements because of their more open terrain. Therefore these results may also extend to snow-free seasons and areas, as reported for black bears in boreal forest (Tigner, Bayne & Boutin, 2014) and cougars in tropical forest (Harmsen et al., 2010).

Among the five body size-diet groups, the observed relative proportions of linear travel movement on SLs in the present study agree with the order expected on the basis of energetic travel efficiency (large predators > large herbivores > mid-sized predators > mid-sized herbivores > small mammals; see Introduction). Furthermore, since body size and diet correlate with metabolic requirements independently of habitat, this pattern of differential benefit, and consequent differences in the amount of SL use for linear travel, could also be expected to occur in non-road linear clearings within other regions and other types of dense vegetation.

Species-specific comparisons of the habitat usage intensity index, between forest and SLs (reflecting the combined outcome of changes in movement and resource use), when considered collectively, provide information about the community-level impacts of these linear clearings. Overall, the winter mammal assemblage within the forest was dominated by medium-sized herbivores (hare, squirrel), with deer (a large herbivore) also common. In contrast, there was a compositional shift within the SL clearings, so that the mammal assemblage was more evenly dominated by large predators (especially coyote), medium herbivores and deer (Figs. 2C and 3C, compare light to dark bars): a change in both species and trophic composition in favor of large predators, and away from medium herbivores. Medium-sized mammals in general (both herbivores and predators) showed a depressed usage intensity in the SL clearings. This may be an avoidance response to the high activity of larger predators in these SLs; single-species research into activity and habitat use by hares (Hodson, Fortin & Belanger, 2010) and marten (Moriarty et al., 2015) in boreal forest reported lower activity in clearings, associated with reduced predation risk.

McKenzie et al. (2012) predicted higher encounter rates between predators and prey in forests with high densities of non-road linear clearings, and thus greater predator efficiency. However, since increased abundance or activity of the large predators may induce various spatial avoidance or other escape responses among different species of prey (which could themselves belong to different trophic levels), the potential community-wide effects of SL-induced predator concentrations are complex. If spatial avoidance occurs, then the prey may suffer from reduced resource access, whereas if other escape mechanisms are used they would be more likely to be more heavily predated; in either case there are likely to be impacts on subsequent survival and reproduction, and ultimate consequences for population dynamics and density. More generally, the large potential for heterogeneous responses to linear infrastructure by different species (Benitez-Lopez, Alkemade & Verweij, 2010) highlights a need for further comparisons of species-specific responses to this and other forms of habitat fragmentation (Gehring & Swihart, 2003).

Conclusions and management implications

Habitat loss and fragmentation present a well-recognized conservation risk for ground-active mammals (Forman, 1995), and may be particularly perilous for mammalian carnivores which have experienced large population and range contractions over the past two centuries (Ripple et al., 2014). However, our results indicate that non-road linear clearings in boreal forest did not reduce most species’ potential movements but instead increased them, especially for large predators and herbivores. Hence, a management focus on simply facilitating crosswise movement in boreal forests containing SLs would be of doubtful value.

Regional extirpations of top mammalian predators (due to various human impacts) have become widespread globally, and have cascading ecosystem consequences that include elevated densities of large herbivores and consequent suppression of vegetation (Ripple & Beschta, 2012). However the boreal forests of the Canadian Rocky Mountains still support a relatively intact and diverse mammal assemblage which includes several species of large carnivore. In the study region, SLs directly occupied 1% of otherwise forested land, while a further 17% of forest was located within 50 m of SL edges (Pattison et al., 2016). This study found that the direct effects of SLs on mammal movements were mostly confined to the relatively small areas of actual linear clearing. However, within these clearings there were large non-barrier impacts on movement and habitat use, with arguably greatest advantage to top predators, associated with an altered trophic structure of the mammal community. When a region is criss-crossed by extensive linear clearing networks, potential population increases in large predators could lead to increased impacts on other species. These consequences may be judged as either desirable or undesirable from a conservation management perspective: for example the benefit to a threatened large carnivore species could be desirable, but such a change in a common predator would be undesirable for a threatened prey species. Understanding the likely outcomes for common herbivores or for vegetation requires further research. Furthermore, these networks of narrow clearings also facilitate human access for activities such as hunting or recreation, with potentially detrimental impacts whose mitigation would require active management. Since this was an observational study we could not draw conclusions with respect to causal relationships due to confounding variables which can only be controlled through randomized experimental designs. We have suggested explanations for our findings but further research is needed to understand the complex effects SLs and other linear clearings have on animal communities.

Supplemental Information

Supplemental Information 1 Raw data.

Click here for additional data file.

Supplemental Information 2 Supplement 1 Calculations.

Example calculations for deer.

Click here for additional data file.

Supplemental Information 3 Table S1. Effect of linear clearings on crosswise movement.

Mean crossing propensity (%, standard error) for forest and seismic line transects (9 surveys) with t-test p-values (N = 14 paired sites).

Click here for additional data file.

Supplemental Information 4 Table S2. Effect of linear clearings on linear travel movement.

Mean linear travel propensity (%, standard error) for forest and seismic line transects (9 surveys) with t-test p-values (N = 14 paired sites).

Click here for additional data file.

Supplemental Information 5 Table S3. Effect of linear clearings on habitat use.

Mean habitat usage intensity (m/km; standard error) for forest and seismic line transects (9 surveys) with t-test p-values (N = 14 paired sites).

Click here for additional data file.

Supplemental Information 6 Mean habitat usage intensity by distance from the nearest seismic line.

Mean habitat usage intensity (m/sub-transect; standard error) for 100 m sub-transects which were located at various distances from the nearest seismic line (0–50 m; 51–150 m and 151–570 m) and means of log transformations of these values which were used to conduct repeated measures analysis of variance (ANOVA) where sites (n = 14) were subjects which were measured repeatedly with track surveys of sub-transects (140 sub-transects). Also reported are repeated measures ANOVA p-values, Greenhouse-Geisser corrected p-values where sphericity was violated (Gg p), results of Shapiro-Wilk tests of residuals for normality (Sh-Wi p), generalized eta squared (ηG2) and partial eta squared (ηp2) which indicate effect size associated with proximity to seismic lines.

Click here for additional data file.

Supplemental Information 7 Crossing propensity post-hoc power analysis.

Crossing propensity results of post-hoc power (1-β) analysis using observed variance, for paired t-tests (n = 14 paired transects) to detect a mean difference between seismic line and forest transects (α = 0.05), of 20%, 50% or 80%, if one existed. Results shown where an effect was not detected.

Click here for additional data file.

Supplemental Information 8 Travel propensity post-hoc power analysis.

Travel propensity results of post-hoc power (1-β) analysis using observed variance, for paired t-tests (n = 14 paired transects) to detect a mean difference between seismic line and forest transects (α = 0.05), of 20%, 50% or 80%, if one existed. Results shown where an effect was not detected.

Click here for additional data file.

Supplemental Information 9 Habitat usage intensity post-hoc power analysis.

Habitat usage intensity results of post-hoc power (1-β) analysis using observed variance, for paired t-tests (n = 14 paired transects) to detect a mean difference between seismic line and forest transects (α = 0.05), of 20%, 50% or 80%, if one existed. Results shown where an effect was not detected.

Click here for additional data file.

Supplemental Information 10 Crossing propensity results of Benjamini-Hochberg procedure to correct for false discover.

Crossing propensity results of Benjamini-Hochberg procedure to correct for false discovery over 19 paired t-tests (n = 14 paired sites) for a significant difference between seismic line and forest transects, with a false discovery rate (FDR) of 0.15. Mammals sorted by unadjusted p-value, bold indicates p < 0.05 and the Benjamini-Hochberg procedure declared a significant result.

Click here for additional data file.

Supplemental Information 11 Travel propensity results of Benjamini-Hochberg procedure to correct for false discovery.

Travel propensity results of Benjamini-Hochberg procedure to correct for false discovery over 19 paired t-tests (n = 14 paired sites) for a significant difference between seismic line and forest transects, with a false discovery rate (FDR) of 0.15. Mammals sorted by unadjusted p-value, bold indicates p < 0.05 and the Benjamini-Hochberg procedure declared a significant result.

Click here for additional data file.

Supplemental Information 12 Habitat usage intensity results of Benjamini-Hochberg procedure to correct for false discovery.

Habitat usage intensity results of Benjamini-Hochberg procedure to correct for false discovery over 19 paired t-tests (n = 14 paired sites) for a significant difference between seismic line and forest transects, with a false discovery rate (FDR) of 0.15. Mammals sorted by unadjusted p-value, bold indicates p < 0.05 and the Benjamini-Hochberg procedure declared a significant result.

Click here for additional data file.

Supplemental Information 13 Habitat usage intensity results of Benjamini-Hochberg procedure to correct for false discovery.

Habitat usage intensity results of Benjamini-Hochberg procedure to correct for false discovery over 19 repeated measures ANOVA tests (n = 140 sub-transects) for a significant difference between sub-transects (100 m in length) located varying distances from seismic lines, with a false discovery rate (FDR) of 0.15. Mammals sorted by unadjusted p-value, bold indicates p < 0.05 and the Benjamini-Hochberg procedure declared a significant result.

Click here for additional data file.

Thanks to L. Stone, D. Soon, G. Gerelus, J. Richards, M. Lord, V. Stockdale, E. Neilson, L. Kemmer, B. Cress, A. Cardinal, C. Davison for help with data collection, and to Pat Dale and Michael Quinn for input into study design.

Additional Information and Declarations

Competing Interests

Author Contributions

Field Study Permissions

Data Availability

The authors declare that they have no competing interests.

Colin A. Pattison conceived and designed the experiments, performed the experiments, analyzed the data, contributed reagents/materials/analysis tools, prepared figures and/or tables.

Carla P. Catterall conceived and designed the experiments, authored or reviewed drafts of the paper.

The following information was supplied relating to field study approvals (i.e., approving body and any reference numbers):

Field data collection was approved by Alberta, Tourism, Parks and Recreation (permit no. 11-014, 12-087).

The following information was supplied regarding data availability:

The raw data is available as a Supplementary File.

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
