# Peer review of "Effects of narrow linear clearings on movement and habitat use in a boreal forest mammal community during winter"

_PeerJ, doi:10.7717/peerj.6504_

## Round 0.1 · original submission · Major Revisions

Both reviewers have provided very thorough reviews suggesting substantial revisions, meeting many of which will improve the clarity and impact of the paper.

The stylistic suggestions are less of a concern, it seems to me, but questions about the appropriateness of terms being used and the analyses are important, and need to be addressed. There is also a need to more clearly detail the methods by which data were collected.

·

Basic reporting

In general, the author used clear and unambiguous, professional English throughout the paper. However, it was not as concise (i.e. short and to the point), as I would have liked to see. Sentences were unnecessarily long.
Abstract: Very interesting findings, but it reads like a discussion. I suggest adding structure to your abstract – 2 lines for intro, methods, results, and discussion. Followed by a conclusion of preferably one line that tells your reader why these findings matter within a global / conservation context.
Structure – the article will benefit from added structure throughout the paper. I have mentioned specific points and suggestions in the sections below
Introduction – Take care to put your most important word / phrase first in a sentence. Most readers do not read entire sentences, but only skim over the most important pieces, which they think should be in the first part of a sentence. For example, the first paragraph of the intro is about fragmentation. Hence, I suggest starting the first sentence of this paragraph with the word – fragmentation. Similarly, the second paragraph is about how type of linear clearings influence animal movement. Hence, I suggest starting this paragraph with ‘Different kinds of linear clearings …’ or something similar.
Last paragraph of the introduction – please start with an overall aim, and then list specific hypotheses / objectives of the study. For example, ‘the overall aim of this study was to determine how non-road linear clearings affect mammalian species of different sizes and trophic levels. The specific hypotheses were 1) that large predators will use linear clearings more than small predators, 2) …’
Results – improve structure with sub-sections addressing each of your original hypotheses listed in the last paragraph of the introduction. Suggestion: 1) SL vs. forest transects: effect on species and different functional groups, 2) SL vs. forest transects: mode of travel (approach vs. cross vs. linear travel), and 3) SL vs. forest transects: effect on snow surface.
Figures and tables – all of the information is there. It just needs a bit of reshuffling.
Table 1 – where it was not possible to discern among species’ tracks within a group (e.g. weasels), and the group comprises of more than one species, list the species as follow: X, Y AND Z. Do this for deer too. Then, refer to the specific group (e.g. weasels and deer) in the main text, and not to the individual species (e.g. termed mule deer and white-tailed deer) within that group. Remember to refer to the supplementary tables in the main text.
Figure 1 – I suggest separating Fig 1a and 1b, and writing a caption for each figure. Key – symbols and words should be in the same line. Figure 1a and 1b – the ‘a’ and ‘b’ should be in the upper left corner of each map. Symbology for Fig 1a – use ArcMap (or a similar package) to change the different elements e.g. seismic lines, pipelines, roads, etc. This will reduce the clutter in the graph. The north arrow should be on the map to which it applies. It cannot go underneath the key, as it does not inform the reader whether it applies to Fig 1a or 1b.
Figure 2 – suggestion for caption: a comparison of mammal movement and habitat use between linear clearings and forest habitat for different species. Add species to the x axes of all graphs. It would have made sense to only label the bottom graph if this figure did not take up the whole page. However, as it is, I have to scroll up and down to see where significant differences lie.
Figure 3 – apply suggested caption for figure 2 to figure 3, but for functional groups. Habitat use intensity – please correct the y-axis. It will be kind of hard to fit in 2000 meters into 1km.
Figure 4 – please change font size of x-axis and y-axis. Add labels (a and b) for moose/elk and shrew.
Raw data shared – shared and it opens without problems
Consistency: Please be consistent in the use of terms, e.g. linear clearings and seismic lines. I realize that seismic lines are one kind of linear clearings, but you have to pick one term and use it throughout the manuscript to prevent confusion.
Terminology: I suggest using the term ‘species’ instead of ‘taxa’. Reason: a taxonomic group can be any group of species or genus or family, depending on your point of interest. It is too general.
Literature – I found it strange that the author did not refer to the papers by Fahrig 2003 (http://www.annualreviews.org/doi/abs/10.1146/annurev.ecolsys.34.011802.132419), Fischer and Lindenmayer 2007 (http://onlinelibrary.wiley.com/doi/10.1111/j.1466-8238.2007.00287.x/full), or Forman 1995 (https://pdfs.semanticscholar.org/1c15/77b4520d6baadb01857877b9714f6858873d.pdf) about the process and effects of fragmentation. In particular, Forman discusses roads (or seismic lines) as dissection of the intact landscape. I suggest placing the article within the context provided by these papers, or providing good reasons why the author chose not to do so.

Experimental design

Pseudo-replication. The size of the study region was ~ 4000 km2, which is 142 km x 28 km. Seismic lines at a density of 2km/km2. There were 14 replicates. Thus, one replicate roughly every 10 km. Large mammals walk further than 10 km in one day. How did you control for the possibility of spotting the same individuals at different localities (i.e. pseudo-replication)?
The research paper falls within the Aims & Scope of the journal – Biological Sciences.
Formulation of research questions needs attention, as mentioned above for the introduction under 'basic reporting'.
Knowledge gap identified: how non-road linear clearings influence mammals. however, I find it difficult to see at a first glance whether studies cited refer specifically to forest habitat, or rather other habitat types. Please clarify, as communities in forest habitat might respond differently to linear clearings than those in grassland.
Reference to species – refer to them first by their common names and then their scientific names (e.g. weasels (Mustela spp.). Thereafter, you may only use the common name.
Lines 80-83: Very valid hypothesis. How did you come to this ‘general observation’? Is it from literature cited earlier, or is it one of the hypotheses that you wish to test? If it is the hypothesis that you wish to test in this paper, move this sentence to the last paragraph and identify it is a hypothesis. For example, ‘we hypothesize that …’
Methods - all important points covered in sufficient detail to allow for replication. However, this section is very long. I think it can be reduced in length by about one third.
Line 110-112: Did you use the same study area as Pattison et al. 2016? If so, state explicitly. In addition, please add a map of your study area. Note that Fig 1a is not a map of your study area, as the caption mentions that it is just an example of a site.

Validity of the findings

Replication – this is my greatest concern for the whole paper, and I am not sure how to deal with it. If the authors could not reliably differentiate among different weasel and deer species (which I completely understand), I do not see how they could have distinguished among individuals of the same species. Therefore, it is possible that they might have counted tracks of the same individual multiple times. This is especially the case, seeing that they worked with non-primitive animals that can have learned behavior. One wide-roaming individual might have contributed significantly to the data recorded.
Statistical analyses – please mention the software used to conduct analyses.
Data distribution - Did you consider data distribution (normal vs. Poisson)? If so, please state explicitly in the text which variables had a parametric and non-parametric distribution. Because you have count data, I would expect your data to have a Poisson distribution. Repeated measures ANOVA is suitable only for parametric data.
All-in-all, I find the data analyses sub-section of the method section to be too simple. Have you considered using Generalized Linear Mixed Models (GLMMs) in R statistical software? It allows you to define the distribution of each variable, incorporate random variables (e.g. kind of forest, days since last snow storm, etc), and then use a standard approach to compare linear clearings vs. forest transect for 1) habitat use, and 2) type of crossing for each species and functional group. Please add references to papers in which statistical methods were developed or tested.
Control sites were sufficient, as it was a paired design, with linear clearings = treatment and forest transect = control / reference group.
Discussion – very well researched, but very long. It is not always clear when you refer to your own work, and when you refer to someone else’s work. Structure your discussion in the same way that you structured your results i.e. same order as your hypotheses. In addition, 1) start each paragraph with your own findings, 2) which contrasts or agrees with person x’s work, and 3) phenomenon Y might explain this finding. Then, you can mention your snow compaction findings, as a potential reason why species (or functional group) Q use linear clearings more often than forest transects.

Additional comments

This paper has potential to contribute to the growing body of knowledge regarding how animals of different sizes use fragmented landscapes, or alternatively – which kinds of animals are most sensitive to fragmentation. In particular, it focuses on non-road linear clearings where animals respond to the shape of a landscape element, rather than to a threat associated with this landscape element (e.g. traffic on roads). As such, it poses a new perspective of dynamic fragmented landscapes, which could be useful for landscape ecology and landscape management.

Reviewer 2 ·

Basic reporting

The authors conducted field work in a presumed remote, difficult to access study area during a time of year (winter) when data collection can be logistically difficult. The study effort was likely high. However, the paper as written does not clearly articulate the study design or validity of results.

- Terms were not defined and inconsistent with the literature.
- References appeared selective, but I did not check these extensively
- Findings and raw data seem unrelated to biologically meaningful results. For instance, there are 36 occasions with >100 red squirrel tracks. Assuming some territoriality, each of these occasions likely represented 1-3 squirrels within a kilometer. Using tracks as an index of use should have careful study designs, standardizing or accounting for time following snowstorms. For instance, Halfpenny (1994) recommended all tracking be conducted within 72 hours of a storm. Also see notes and suggestions within general comments.

Experimental design

- Experimental design and site selection was not explained.
- Research question defined, but data not collected to a high technical standard as explained
- Methods do not contain enough detail to replicate study (see notes within general comments)

Validity of the findings

- Data does not appear statistically sound.
- Conclusions are not consistently linked to data or results.

Additional comments

Introduction in general:
There are several words that have meanings in the literature that are either odd or incorrect throughout the manuscript.
I’d either further explain, or tone down the functional group concept for which most of the paper is based on. You lumped species by size and whether or not they were plant vs meat eating essentially. In each category, those classes should not behaviorally similar. For instance, large predator (grizzly bear, cougar, gray wolf, coyote, lynx) encompasses every strategy, e.g., cougars are sit and wait predators, wolves are social pack animals, lynx are solitary specialists. You would expect each species, based on the literature, to use seismic lines differently. You’re ignoring behavior (and the “functional” group) and testing whether species by size and diet class is more influential. You explain this in paragraph 66 stating that functional groups might not be the best approach and focus your paper accordingly BUT you use the term functional group in your abstract (line 27), suggesting that these are functional groups – they are not.
You often use the word habitat to mean vegetation or patch. Habitat could be defined as:
• a distinctive set of physical environmental factors that a species uses for its survival and reproduction that produce occupancy (Hall et al., 1997; Krausman and Morrison, 2016)
• “the resources and conditions present in an area that produce occupancy, including survival and reproduction, by a given organism. This includes more than vegetation structure, but the sum of specific resources” (Krausman, 1999)
• “the set of resources necessary to support a population over space and through time” (McComb, 2007)

Here, there is no information on survival, reproduction, long term resources, nor do the data ever have this capacity. As such, the term habitat seems inappropriate. The term “habitats” is nonsensical. The term functional habitat (e.g., line 29) is confusing and even more incorrect, confusing the reader with the functional group phrase (neither are functional or defined as such). In short, these diverse animals are not using habitat (e.g., seismic lines) unless you can convincingly determine that each species is using the area for increased reproduction or survival – movement is different. We can swim across a lake, traveling entirely across or entering briefly and exiting. Neither event would make the lake human habitat. Consider forest, patch (and define patch), or the actual description of what you’re describing.
You are sampling animal movement trajectories during winter in areas that differ in forest cover and configuration.
Specific comments:
Line 23: First line – “internal fragmentation” does not make sense. Fragmentation obviously increases the number of patches and reduces the amount of “core” area, but here it is not defined. Is “internal” something that is >5km or 50m from any disruptive feature?
Line 57: “Open habitats” is an incorrect use of the term habitat (please see Krausman and Morrison 2016, Hall et al. 1997). Similar issue line 62, 93, 146, .

Line 80: Define mammal functional groups. Is large predator a functional group? This premise is unclear within the paragraph and throughout.
Lines 116-119: This would be a good location to describe a seismic line for the reader – how wide? How long?
Lines 122-130: How were the 14 replicates selected? Were these a random sample, stratified random, or selected? How did you account for differences in seismic width?
Lines 132-135: Here, you suggest that sites were maximizing spatial dispersion (ave = 6km distance, maximum = 12km). However, for all large and mid-sized carnivores and larger herbivores it would be easily conceivable for the same individuals to occupy multiple transects. I’d consider a new Figure 1 with panels – one that shows the locations of these sites spatially and a demonstration of how the surveys were conducted.
Line 135: what is original coniferous forest cover?
Line 136-9: Can you more clearly define what you mean by average forest cover affecting sites? I’m assuming that you mean your sampling estimate of forest cover. Even so, it’s not clear why you would mention this and what it’s in reference to. Are 2 sites with linear seismic lines covered in 6m high trees and thus are no longer open?
Lines 143-146: I think I can understand what you’re saying here, but it’s not clear. You have 2 treatments: open seismic line transects and perpendicular forested transects, presumably with fairly closed canopy which is why you mentioned 98% canopy cover (line 138). I’d strongly recommending spending more time on the study design and purpose, clarifying your two treatments with clear and consistent diction. For instance, you can set up
- Where sites were chosen
- How sites were stratified – are all similar, or are you accounting for width and type (e.g., you mentioned seismic lines with propane)
- How did you pick your direction and placement of the perpendicular forested transects? Are these always central, creating a “T”? Do they always stem to the west like shown in figure 1?
- What were the conditions of the open and forested transects
o Canopy cover
o Tree size
o Tree height
Lines 147-152: Can you further explain your snowtracking methods? Did you use suggestions by Halfpenny (e.g., Halfpenny 1994)? How was the snow measured? Was the survey medium the same during all 6 surveys? What was the duration between storms? Do you expect any difference in detectability between surveys?
- Since you did repeated surveys within a season, were they at the same locations? For instance, did you resurvey the 14 sites EACH time (9 surveys x 14 locations = 126 tracking opportunities) or did you go to some of the sites on each survey?
o If you went to different sites each survey, how are you accounting for differences in timing?
o If you went to the same sites each time, could you use an occupancy model to verify that your detectability was high?
Line 154: How was an observer trained?
Line 158-160: This might be better clarified in the introduction with citations why you chose these categories. Can you use these data to compare to other studies and test their predictions? It seemed like the introduction was moving towards larger hypotheses, so it would be helpful to explain why and how. It’s also odd – why wouldn’t large herbivores also be >10kg to match the predators? Why are mid-sized predators and herbivores different sizes? Seems like you had a plan… but explain for the reader.
Lines 163-168: Explain to the reader why 10m? Why is this relevant? You’re using the same methods for diverse species but that seems odd. 10m to a shrew is much different than 10m to a wolf. 10m to a wolf is 2-3 strides, perhaps 10 seconds of the animal’s day. Does it ecologically matter to a wolf whether during that moment it was moving along or across? It might, and it would be great to explain how you are accounting for size (aside from clumping groups by size). In pondering, the way that you would be able to increase meaning is by increasing sample size. If the animals are similar across all 14 sites, or across 100 sites… then you have a pattern. Here, it’s not clear why you are collecting data and what pattern you expect to see.
Line 179: here is the first time you mention the width of a seismic line. That is WAY too late and offhanded for a paper founded on the study of seismic lines. Incorporate in the introduction or early methods.
Lines 176-182: These metrics seem mostly logical, but also created. Are there no movement metrics in the literature that could be compared or used? Did you take path measurements to calculate turn angles for instance? I’d justify these methods a bit more. Because they are logical, I think you can make a case. I’d also write in the active voice. Instead of “crossing propensity was calculated as” say “we estimated an index of crossing propensity as”
Line 185: clearfell forest harvesting is not defined or cited. It’s also unclear why harvesting reduced your transect length? Do you mean you created a forest transect and during the study the forest was cut – thereby decreasing your ability to survey?
Line 189: Can you define (earlier) your fundamental unit of measure? Is a track a footprint or a path?
Line 190: Ah… snow measurements. I’d put this section earlier during the tracking portion. This is great, but it’s not clear how and why this is used in the analyses.
Line 199: How would a repeated measures ANOVA tell you differences by taxa? Wouldn’t it merely show a difference in time and not by subject (species)? Or did you do this separately by species?
Line 201 – this suggests you resurveyed each of the 14 sites each year, but your methods suggest you had 2 surveys in year 1, 4 in year 2, and 3 in year 3.
Line 207: Paired t-tests seem okay if you have a single response variable for a single paired question, but you likely have correlations between species within site at between sites. I suppose you can do a multiple comparison correction (e.g., Tukey, Bonferroni) but I’m assuming that a generalized linear mixed model would be more appropriate. Since you’re not interested in either survey time or site, both could be random effects. Your fixed effects could be as simple as vegetation type (1 degree of freedom), snow depth, and species. You have 3 response variables. You could also test for assumptions of heteroscedasticity and ensure your results are valid. Recall for any of these tests – your sample size is 14.
Line 218: I’m confused here. You traveled a total of 1 kilometer, but you’re measuring distance up to 570m. Please explain in the methods as I’m assuming there was additional landscape variables calculated? Also, your sample size is still 14, correct? I’m not clear how some distances would have a sample of 36 and others with 60 with the further distances having increased samples. Perhaps an additional figure and description of how sites were chosen should be included. If you do have this many crossings, then you’re likely not accounting for landscape level variation, which has been significantly influential in other research (e.g., the percentage of patches, the size and core area, etc).
Line 225: You’re differentiating between voles, mice, and shrews… I’d either report how you can do so in the methods or lump your small mammals.
Line 233: I’m assuming based on your table, that your 95% confidence in the figures are based on the number of tracks and not the number of sites. If so, that would be pseudoreplication and inappropriate. Please define in the methods and change accordingly.
I stopped line edit suggestions here due to a lack of time and inability to decipher results.
Paragraph 270: ANOVAs will provide information as to whether one group is significantly more or less – but not which group. Also, see evidence and study designs to assess small mammal use e.g., Rosenberg, D.K., Noon, B.R., Meslow, E.C., 1997. Biological corridors: form, function, and efficacy. BioScience, 677-687.
Paragraph 280: Your statement in line 283 about statistical power is concerning. Your sample size is 14 and you do not demonstrate that those locations were random – this is a very small study to be making such statements. Please reassess.
Paragraph 294: Your study did show an effect on shrew movement (assuming you can differentiate shrew tracks accurately). This paragraph seems to stray from your results.


Figure 1: See notes within. I’d have a study area map, then inset with a transect set up and the designations for crossing/linear travel. Your figure 1a isn’t exactly clear. Can you use a different icon for roads? Are seismic lines and non-forest different? If so – as expected if created from differing processes, I’d suggest other colors/textures. Lastly, on your legend, please put the symbol to the left of the feature.
Figure 2,3: I’d change the forest to be grey and the seismic lines (open) to be white. It’s counter intuitive currently.
Figure 2, 3: define each response. What’s a habitat usage intensity? Also, see note above – this is an incorrect use of the word habitat by definition. I suggest changing to vegetation or seismic line throughout. This paper doesn’t address habitat

---

## Round 0.2 · Minor Revisions

In general a good job has been done thoroughly addressing the concerns raised by the reviewers, but there are some outstanding issues, particularly with explaining the nature of the analyses, that need to be addressed.

Reviewer 2 ·

Basic reporting

Writing is generally okay, but some of the definitions are not common or correct in ecological literature. Specifically, I am still concerned about the use of "habitat" and "functional groups". Mostly, I suggest the word habitat is removed as no formal analyses of predicted habitat were completed nor do the authors demonstrate data or capacity to evaluate predicted habitat. Functional groups of mammals were classified by weight, ignoring ecological aspects of each species. My suggestion is to remove the entire mention of functional groups and the associated figure. The information will be more transparent and biologically meaningful without that distinction.

The introduction highlights novelty of this study evaluating communities within non-road linear features, which may be a novel assertion for seismic lines. Authors fail to consider the breadth of information/literature from utility lines, which are also non-road linear features. Otherwise, there are only minor changes suggested. I was only able to read through the methods and results. I suggest major revisions before another review.

Raw data are shared, but I was unable to replicate any of the information and tables and it's unclear that the analyses were appropriate. It is unclear how the raw data were used to create the data summaries (elaborations in the validity of findings section).

Experimental design

The paired design between seismic lines and forest interior is helpful, as the authors are taking into account local conditions over time. This work comprises a huge field effort but, unlike the authors argue, is modest to small in terms of sample size.

First, data were not randomly collected. This is an observational study and results can be applied to these areas during the time of the study, as verified by the authors in their response: comment 47 "Regarding site selection, some revisions to the text have been made, but we have avoided giving the excessive rationale now provided below, because these types of issues are very common in ecological studies that use multiple sites, and it is not usual practice to attempt to describe them in detail. Sites were not selected randomly or systematically in space because random selection without attention to the distribution of possible factors that affect habitat use can result in confounding between independent variables of interest with some other related but unknown factor (Hurlbert 1984; Fahrig & Rytwinski 2009)." Non-random selection is common in ecology due to logistics and safety and this should be made more clear in the methods (paragraph beginning line 128).

I'm curious about the response to comment 58. "Finally, we reject the premise that increasing sample size would address this issue." I suggest using your observed variation in data between sites and conducting a power analysis to better understand if you have enough data to draw conclusions.

Validity of the findings

Raw data are shared, but I was unable to replicate any of the information and tables. It's extremely unclear that the analyses were appropriate. It's unclear how a particular value (which should be within 1/10 of a segment) was used to derive percentages in the figures or supplemental tables.

Based on the methods, I presume the data include the number of tracks by species within 100m. Authors provide the site (n=14), the treatment type (forest, seismic line), the portion of the transect within 100m segments, the type of movement activity (crossing, approach, linear travel along transect), and a count for each species. I assume the distance the number of meters traveled. In this case, it would be helpful to know results by year and time.

Linear distance is unclear and how a measurement is represented by the other data. There should be an opportunity to clearly summarize use and intensity.
- As an example, within a 100m segment of a portion of a site, there was a linear distance of 4162m for coyote (site = 14, type = Seismic, subtransect = 1). Does that mean within a single 100m segment one or more coyotes traveled >4km during 3 winters? For this same region there was 6 crossings and 0 approaches. The methods suggest that there's a net longitudinal displacement calculation, but this is not shown.

Because I was unable to understand the linear values, I went through the raw data more. Some clarity is needed. As an example,
Site = 1
Type = Forest
Subtransect = 1

For deer, you record 1 approach, but you have 10 crossings. Does that mean a single deer approached and crossed 10 times within 100m?

I attempted to repeat your formula for crossing propensity for hare. By transect, I summed your raw data in a pivot table by each treatment. Then, I calculated your cross propensity using your formula and I think I was able to obtain similar estimates to Figure 2a. My average crossing propensity appeared similar, but not the same as yours with 96 and 87% for forest and seismic lines. My standard error was low, suggesting a difference. A paired t-test in excel suggested P<0.001, similar to your figure. Nonetheless, here your data is suggesting (I think) the number of footprints differed which doesn't make any sense to me. If you assume a hare has a home range of 100m (or even half that), then you would be more interested in a count along the transect and a frequency along transects. If you do the same test with the count data (Excel Sheet from your raw data, Hare Cross Tab) then there is no difference between hare crossing propensity between the forest and seismic lines (P = 0.41). This is an example of why the reader cannot directly take your thought process (as developed in the introduction), methods as written, raw data and come to similar conclusions.

I would strongly suggest that the authors walk the reader through, step by step how data were analyzed either in the main body or a supplemental guide. Further, if possible, I would use an R script or a word document to describe how data were altered for each summary and analysis.

Further, there may be some evidence that the authors have looked at the data more thoroughly. In the response comment 26, authors suggest "Friedman tests were conducted where residual plots indicated a non-parametric distribution." After describing how data were used, I would include these preliminary plots as supplemental information.

Additional comments

There are some scientific opportunities within this paper and I applaud the authors for this field effort. I'm still very unclear about the validity of the analyses, summaries, and results. Before I can understand the discussion and context of these findings in the literature, the analytical methods and results would need to be clarified.

In general, I don't feel as if you respected the suggestions by your reviewers and this made the paper difficult for me to approach. It seemed like you spent time providing arguments, but not I'm not convinced that your arguments were scientifically valid. For instance, the other reviewer suggested reviewing and summarizing the data differently (comment 27) and provided opportunities to do so. Your response that the analyses were appropriate for the research questions seem incorrect, especially since it's very unclear what you actually did with your data. In some (like comment 42 regarding habitat), you provide citations that cited Forman 1995, a book on land mosaics ignoring the wealth of information on the term and correct usage for the word habitat. Although I could spend time working through the paper pointing out many discrepancies, I highlight only a few.

You may have an interesting data set, but it's not clear how you're reporting results and why you're summarizing data as described. See some of my comments within the pdf.

Annotated reviews are not available for download in order to protect the identity of reviewers who chose to remain anonymous.

---

## Round 0.3 · Major Revisions

Thank you for carefully considering both rounds of reviews and making amendments to accommodate them - I think this has made the paper both stronger and clearer, but there are remaining issues that one of the reviewers has serious concerns about.
Please continue your diligent efforts to address those concerns in the next revision.

Reviewer 2 ·

Basic reporting

The authors improved the manuscript for this version considerably. I found the introduction to read better and provide additional clarity for the reader.

- Habitat: the authors revised much of the introduction to remove the word habitat with a vegetation class. That was appreciated and the manuscript reads more accurately.

- Functional groups: I would suggest replacing the term "functional group" with their own category as used in the caption for Figure 3: "groups based on body
size-diet combinations", adding within the abstract/introduction/methods hereafter "size-diet groups". To me, "functional group" seems like a key word search to allow the impression of broader impacts. I disagree that a function group can be condensed by weight and diet. With an extreme example, a great horned owl, large mouth bass, and house cat could be considered a "functional group" within the author's use (size/predators), but that's not correct due to differences in behavior and obvious differences in mobility. Here, these species also differ in their movement and how they would perceive linear features.

I'm not a fan of Wikipedia for definitions, but this is the first paragraph, emphasizing a functional group with characteristics linked to a niche and fitness [Accessed 31 Oct 2018]:

"A functional group is merely a set of species, or collection of organisms, that share alike characteristics within a community. Ideally, the lifeforms would perform equivalent tasks based on domain forces, rather than a common ancestor or evolutionary relationship. This could potentially lead to analogous structures that overrule the possibility of homology. More specifically, these beings produce resembling effects to external factors of an inhabiting system.[1] Due to the fact that a majority of these creatures share an ecological niche, it is practical to assume they require similar structures in order to achieve the greatest amount of fitness. This refers to such as the ability to successfully reproduce to create offspring, and furthermore sustain life by avoiding alike predators and sharing meals."

Experimental design

I'm concerned that the authors are arguing but not thoughtfully addressing potential issues within their response:

For example Comment 8, Reviewer 2 rebuttal: "We stand by our original statement and reject the premise that increasing sample size would address the comment (above) because this was not a research question posed by this study. With respect to the current comment which refers to “conducting a power analysis to better understand if you have enough data to draw conclusions.” We believe that this is unnecessary for the following reasons:
1) “Statistical power is a measure of our confidence that we would have detected an important effect if one existed.” (Quinn & Keough 2002). Since we detected an effect for several taxa and functional groups, power was sufficient in those cases.
2) With respect to non-significant results, we accept that the reviewer may be concerned about our conclusion that mammal movements across transects were either unaffected by seismic line clearings or restricted only slightly. However, we have included figures, mean values, measures of variance and statistical test results in the current manuscript. It has been demonstrated that p-values and power are inversely related and that low p-values mean higher power and high p-values mean lower power (Thomas 1997; Hoenig & Heisey 2001). Since we have included p-values, the addition of post hoc power calculations would not add new information to the manuscript.
3) In addition, the manuscript’s figures and supplemental tables provide the necessary information for readers to draw conclusions. For example, Figures 2a and 3a show that there was very little difference in crossing propensity between seismic line and forest transects (i.e., a small effect size). Therefore, irrespective of the power of the statistical test, the figure shows the reader that mammal movements were restricted only slightly, or not at all because a) there was little difference in crossing propensity between the 2 types of transects (i.e. forest versus seismic line) and b) seismic line crossing propensity was high, as 89% of tracks approaching seismic lines, subsequently crossed them. A power analysis would not change this finding."

Certainly, few studies have the fiscal capacity to achieve a desired sample size. Here, the authors failed to conduct a power analysis and its not clear if they have data of sufficient capacity for proper interpretation. This is an observational study, and therefore descriptive. Given what I've read, I'm hesitant to believe the first statement in their discussion is appropriate:

"Most taxa in this boreal region showed no difference in crossing rates of seismic line clearings compared with forested control transects: these non-road linear forest clearings did not create strong barriers to movement."

This statement could have profound implications for conservation and the data are not robust (in my opinion) to clearly state whether or not linear clearings were barriers. Track data during winter and the analyses within this manuscript allow for hypothesis generation. There are potential implications from these data, but the evidence is not sufficient as the manuscript (and supplemental materials) are written.

Please see my comment within the validity of the findings. Using the argument that because the authors report p-values, having sufficient statistical power is unnecessary is incorrect.

Validity of the findings

- Supplemental tables: The authors negate many of the concerns in the last review by providing additional supplemental tables. "We have also created a supplement to describe the process used to calculate the results shown in Figures 2 and 3 and Supplementary Tables 1, 2 and 3 and to address concerns which were raised regarding the nature of the analysis (points 5, 9-14, 16, 17 and 19)." [1st comment to the editor.]

The authors suggest that Supplemental table 1 addresses concerns on 11 occasions.

These supplemental tables are NOT providing me with reassurance that data were properly assessed.

First, the Supplemental tables do not address the reviewer concerns or it is unclear how they do so.

Second, the tables provide t-tests for all species, groups, and they test each possible transformation (Footnote: "Transformations were applied to data sets prior to conducting paired t-tests. 2nd indicates square root transform, 3rd is cubed root, etc. Log applied as log(x+1)."). The P-values require a Bonferroni correction or a correction for false discovery like the Benjamini-Hochberg procedure.

From a free online bio-statistical handbook [http://www.biostathandbook.com/multiplecomparisons.html, Accessed 31 Oct 2018]: "Any time you reject a null hypothesis because a P value is less than your critical value, it's possible that you're wrong; the null hypothesis might really be true, and your significant result might be due to chance. A P value of 0.05 means that there's a 5% chance of getting your observed result, if the null hypothesis were true. It does not mean that there's a 5% chance that the null hypothesis is true.

For example, if you do 100 statistical tests, and for all of them the null hypothesis is actually true, you'd expect about 5 of the tests to be significant at the P<0.05 level, just due to chance. In that case, you'd have about 5 statistically significant results, all of which were false positives. The cost, in time, effort and perhaps money, could be quite high if you based important conclusions on these false positives, and it would at least be embarrassing for you once other people did further research and found that you'd been mistaken.

This problem, that when you do multiple statistical tests, some fraction will be false positives."

Because the authors are interpreting their data within a scientific publication, it seems that each piece of evidence should have assumptions clearly stated and the analysis most fitting to the data. These data could be incorporated within a generalized linear mixed model or tests could be completed with a multiple comparisons adjustment.

- Raw data and repeatability: there was no apparent effort made to refine data or allow readers to repeat calculations as per my prior review comments. Please reformat the supplemental materials to address prior concerns.

- Figure 4: This is extremely confusing: "Effect of distance from seismic line (SL) clearings on habitat usage intensity within forest transects, for (a) moose and (b) shrew which were measured repeatedly in sub-transects located various distances from seismic lines (n=36, 44, 60 for 0-50 m, 51-150 m, 151-570 m respectively) in 14 sites (subjects); results of ANOVA test shown (+P<0.1, *P < 0.05)."

Is the graph a mean and confidence interval, or output from an ANOVA? Please clarify. An this analysis of variance test would only show whether or not there was a difference between populations (I'm assuming distances). Further, an ANOVA is for independent data that is normally distributed and has equal variance. In the caption, it's clearly stated that these data are not independent; there is no mention of a Levene's test for equality... I'm fairly certain this is an inappropriate test.

Additional comments

I apologize for the extended time of this review. I have recently transitioned jobs and am attempting to juggle two positions simultaneously.

I believe this study required a remarkable amount of field effort. I would consider these and prior comments carefully. I don't feel that many of my original concerns were addressed. I am disappointed to be in a place of argument rather than collaborative improvement. I feel this manuscript could benefit from transparency, and thoughtful interpretations with clearly stated caveats, and ideally the assistance of a trained statistician. The discussion, as written, leans towards confident opinions and I am skeptical that your data and evidence are appropriate to make strong claims. There is great opportunity reporting your results in a peer reviewed journal and I applaud your tenacity. Please consider revisiting this paper, ensuring the evidence is correctly portrayed and your interpretation matches the weight of the evidence. Also consider the wealth of literature moving away from p-values and stressing the importance of effect size, an issue not addressed within.

---

## Round 0.4 · accepted · Accept

Thank you for carefully incorporating the suggestions of the reviewers.
Reviewer 2 was not satisfied with revisions that had been made, but my judgement was that you have a valuable (and difficult to collect) data set that have been adequately reported, and appropriately interpreted, and so I have accepted the MS.

#